

# MOVEIM v1.0: Development of a bottom-up motor vehicular emission inventories for the urban area of Manaus in central Amazon rainforest.

Paulo Ricardo Teixeira[1,3], Saulo Ribeiro de Freitas[5], Francis Wagner Correia[2], Antonio Ocimar Manzi[4]

[1] Post-graduate Program in Climate and Environment, CLIAMB, INPA/UEA, Av. André Araújo, Manaus, Amazonas, Brazil
[2] State University of Amazonas, Meteorology Department, Manaus, Amazonas, Brazil.
[3] National Institute of Amazonian Research, Large-Scale Biosphere-Atmosphere Experiment in Amazonia, Manaus, Amazonas, Brazil
[4] National Institute of Space Research, Center for Weather Forecasting and Climate Research, Cachoeira Paulista, São Paulo, Brazil
[5] Universities Space Research Association and NASA Goddard Space Flight Center, Greenbelt, MD, USA

*Correspondence to*: Paulo R. Teixeira (paulo.ricardo.teixeira@gmail.com), Saulo R. Freitas (saulo.r.freitas@nasa.gov) and Francis W. da Silva Correa (francis.wagner70@gmail.com).

**Abstract**. Emissions of gases and particulates in urban areas are associated with a mixture of various sources, both natural and anthropogenic. Understanding and quantifying these emissions is necessary in studies of climate change, local air pollution issues and weather modification. Studies have highlighted that the transport sector is key to closing the world's emissions gap. Vehicles contribute substantially with the emission of carbon dioxide ($CO_2$), carbon monoxide (CO), nitrogen oxides ($NO_x$), non-methane hydrocarbon (NMHC), particulate matter (PM), methane ($CH_4$), hydrofluorocarbon (HFC) and nitrous oxide ($N_2O$). Several studies show that vehicle emission inventories are an important approach to providing a baseline estimate of on-road emissions in several scales, mainly in urban areas. This approach is essential to areas with incomplete or non-existent monitoring networks as well as for air quality models. Conversely, the direct downscale of global emission inventories in chemical transport and air quality models may not be able to reproduce the observed evolution of atmospheric pollution processes at finer spatial scales. To address this caveat, we developed a bottom-up vehicular emission inventory along the 258 main traffic routes from Manaus, based on local vehicle fleet data and emission factors (EFs). The results show that the light vehicles are responsible for the largest fraction of the pollutants, contributing 2.6, 0.87, 0.32, 0.03, 456 and 0.8 ton/h of CO, $NO_x$, $CH_4$, PM, $CO_2$ and NMHC, respectively. Including the emissions of motorcycles, buses and trucks, our total estimation of the emissions is 4.1, 1.0, 0.37, 0.07, 63.5 and 2.56 ton/h, respectively. We also noted that light vehicles accounted for about 62.8%, 84.7%, 87.9%, 45.1%, 71.8%, and 33.9% and motorcycles in the order of 32.3 %, 6.5 %, 12.1 %, 6.2 %, 14.8 %, 8.7 %, respectively. Nevertheless, we can highlight the bus emissions which are around 35.7% and 45.3 % for NMHC and PM. Our results indicate a better distribution over the domain reflecting the influences of standard behavior of traffic distribution per vehicle category. Finally, this inventory provides more detailed information to improving the current understanding of how vehicle emissions contribute to the ambient pollutant concentrations in Manaus and their impacts on regional climate changes. This work will also contribute to improved air quality numerical simulations, provide more accurate scenarios for policymakers and regulatory agencies to develop strategies for controlling the vehicular emissions, and, consequently, mitigate associated impacts on local and regional scales of the Amazon ecosystems.

**Key words:** Vehicle emission inventories, bottom-up approach, urban air pollution, Amazon forest



## 1 Introduction

Since the pre-industrial era, the climate changes driven by anthropogenic emissions are one of the most pressing challenges faced by human development. Researchers have highlighted the effects of human activities on air quality, terrestrial ecosystems and
global climate changes (Ramanathan et al. 2001; IPCC, 2014; Ashfold et al., 2015). The fast economic and population growth with a consequential increase in the number of vehicles has contributed significantly to the greenhouse gas (GHG) emissions and other and pollutants, therefore, affect the climate in global scale (Chapman, 2007; Stanley et al., 2011) besides bringing several adverse consequences on human health (Afroz et al., 2003; Abe et al., 2016; Scovronick et al., 2016; Yoshizaki et al., 2017). The GHG emissions associated with the transport sector have increased at a faster rate. Kahn et al. (2007) shown that in 2004, the
transportation sector was responsible for 23% of the world's GHG emissions with about 75% coming from road vehicles. Recent studies suggested that of the atmospheric emissions the vehicle fleet is globally responsible for 30% of $NO_x$, 25% of PM2.5, 54% of CO and 14% of CO2 (Vasconcellos, 2006, Sokhi, 2011; Karangulian et al., 2015). Additionally, Sims et al (2014) shown that in 2010 the GHG emissions from the transport sector continued to increase at a faster rate more than any other sector with 7.0 Gt $CO_{2eq}$ this represents more than the doubled since 1970 (2.9 Gt $CO_{2eq}$), 53% of these increase are from passenger modes and the
47% are freight mode, totaling 5 Gt $CO_{2eq}$. The authors consider that more knowledge is needed about the worldwide potential for GHG emission reduction from the transport sector and that potential reduction is much more certain for passenger modes. During the United Nations Conference of Parties 21 (UN COP21) Climate Change Conference, it was highlighted that the transportation sector is key to closing the world's emissions gap (Ebinger et al., 2015). The world's emissions gap represents the difference between the emissions levels that countries have pledged to achieve under the Paris Agreement to combat climate
change. The main goal is to hold the increase in the global average temperature to below 2.0°C, compared to pre-industrial levels and consistent with the global effect of the Intended Nationally Determined Contributions (INDCs) (Rogelj et al., 2016; UNEP, 2017).

However, an air pollution monitoring network in many urban areas is either unavailable or inadequate. Alves et al (2014) suggested that in Brazil only 1.7 % of the cities have an air pollution monitoring network, which represents 1.3 stations per 1
million inhabitants. These numbers are considerably lower than those in the U.S. (16 stations per 1 million inhabitants) and Europe (14.8 stations per 1 million inhabitants). Thus, vehicle emission inventories are a simple and needed approach that would provide a baseline estimate of on-road emissions on several scales. This information is essential to areas with incomplete or non-existent monitoring networks (Nagpure and Gurjar, 2012) and to constrain pollutants surface fluxes for numerical air quality models (Coelho et al., 2014; Lozhkina and Lozhkin, 2015; Vara-Vela et al., 2016). As a means to minimize this gap in
monitoring network, some studies suggested in general, two different approaches: top-down and bottom-up. The top-down approach uses values of annual emissions assessed at national levels; these emissions are spatially disaggregated at different levels by statistical indexes (e.g. population density, average number of trips, etc.). However, the 'bottom-up' approach is more accurate and uses data at local and municipal levels (e.g., locally measured emission factors, vehicular activity, fuel consumption, traffic characteristics, fleet characteristics, length of road, etc.). In several studies, both methods are utilized for
improving the accuracy of the emissions calculations applying a downscaling national emission, therefore disaggregating atmospheric emissions from the national scale to a city-scale (Kuenen et al., 2014; Janssens-Maenhout et al., 2015; Trombetti et al., 2017). In addition, Dios et al. (2012) suggest combining top-down and bottom-up methodologies for high-resolution inventories. However, despite efforts Zhou et al (2017) suggested that direct application of downscaled global emission inventories in chemical transport and air quality models, may not be a good solution to reproduce the real evolution of
atmospheric pollution processes at smaller spatial scales suggesting that the application of downscaling produces emission overestimates.



Recent efforts have been made to develop vehicle emission inventories for several urban areas (Song et al., 2006; Bellasio et al., 2007; Jing et al., 2016; Andrade et al., 2010; Alonso et al., 2010; Venegas et al., 2011; Zhou et al., 2014). Particularly, a few years ago, in South America studies have reported information on vehicle emission inventories on a national scale or for specific
cities. Gallardo et al. (2012) presented an evaluation of vehicle emission inventories for CO and NO$_x$ at Bogotá (Colombia), Buenos Aires (Argentina), Santiago (Chile) and São Paulo (Brazil). Puliafito et al. (2017), Puliafito et al. (2015), Venegas et al. (2011) have suggested a spatial disaggregated emission inventory in high resolution for Argentina. Alonso et al. (2010) developed an urban emissions inventory for South America based on the analysis and aggregation of local inventories of nine megacities, using socio-economic data of the region and correlation between vehicle density and mobile source emissions. This
information was extrapolated geographically and distributed using a methodology that delimits urban areas using high-resolution satellite data and finally was integrated with worldwide emissions databases. In Brazil, several studies reported information on vehicle emission inventories in specifics cities such as Federal District (Réquia Júnior et al., 2015); Rio de Janeiro (Duarte et al., 2013); São Paulo (Andrade et al., 2004; Vivanco and Andrade, 2006; Martins et al., 2006; Gallardo et al., 2012); Campinas (Ueda and Tomaz, 2011). However, despite the efforts to implementer an air pollution monitoring network in national scale the
North, Northeast and Midwest regions of Brazil are characterized do not have emission-specific inventories at local and municipal levels.

In Brazil, in the North Region located in middle of the world's largest rainforest is the city of Manaus, which stands out for being a large urban area surrounded by primary tropical forests (Martin et al., 2016; Cecchini et al., 2016). Given such an environment, it is key to study the impact that Manaus has on atmospheric conditions. Studies have shown that the Amazon
rainforest is sensitive to the variability and changes in the climate system due to both natural variations and anthropogenic actions, such as the increase in the concentration of GHG and aerosols in the atmosphere and changes in land use and land cover (LULC) (Fearnside et al., 2002; Artaxo et al., 2006; Betts et al., 2008; de Souza e Alvalá, 2014; Marengo e Espinoza, 2015). Previous studies of emissions have shown the effects of anthropogenic impact on the Amazon basin generally focused on biomass burning-related occasions (Artaxo et al., 2002; Roberts et al., 2003; Andreae et al., 2004; Freud et al., 2008; Martins and
Silva Dias, 2009; Artaxo et al., 2013). However, over the last years, several researchers have highlighted the effects of the Manaus plume pollution on the chemical properties of the local and remote atmosphere (Kuhn et al., 2010; Trebs et al., 2012; Rizzo et al., 2013; Bateman et al., 2017). During 2014-2015 the experiment Observations and Modeling of the Green Ocean Amazon (GoAmazon2014/5) was performed in the Manaus metropolitan area, the objective being to understand and quantify the impact of the pollution plume from the Manaus urban area on the complex interactions among vegetation, atmospheric
chemistry, and aerosol production as well as their connections to aerosols, clouds, and precipitation (Martin et al., 2016). As part of the GoAmazon2014/5 Experiment, the most recent results indicate that Manaus pollution plume impacts in the microphysical properties (e.g., smaller effective diameters and higher droplet number concentrations) of warm-phase clouds during wet season (e.g., Cecchini et al., 2016). In addition, de Sá et al. (2017) suggested that urban pollution is responsible for the increased emissions of nitrogen oxides and decreased particulate matter (PM) compared to background conditions at Amazon rainforest.
The authors indicate that this results corroborating with some futures scenarios of Amazonian economic development. In this context, the possible change of PM production would imply in alterations on air quality and regional climate.

Therefore, given such an environment it is key to quantify mobile sources emissions of Manaus city. Specifically, the purpose of this paper is to develop a vehicular emission inventory along the main traffic routes based on vehicle activity data and local emission factors using a bottom-up methodology.



## 2 Methodology and data

### 2.1 Brief description of the study area

Manaus is situated at the confluence of the Negro and Solimões Rivers, more precisely between the parallels 2°55'00" and 3°10'00" south and the meridians 59°52'30" and 60°07'30" west, occupying a total area of 377 km² (Figure 1). With a population of more than 2 million is 7[th] most populous city of Brazil (IBGE, 2015) being responsible for about 80% of the economic activity of the state of Amazonas. With quick growth, with a rate average in the order of 7% during 2010-2016, Manaus has seen the increase in the vehicle fleet as one of the major sources of air pollution. The motor vehicle census data from the National Transit Department (DENATRAN), show there were in 2010 around 453,000 vehicles registered in Manaus; today, it is estimated at ~713,000 vehicles (78.6 % of the vehicles of the state of Amazonas), meaning a 57.6% increase. At an annual growth rate of 7%, by 2020 the vehicle fleet could reach 1 million. The Manaus fleet is relatively young age and predominantly consisting by flexible-fuel vehicles (blend between gasohol and hydrated ethanol); Medeiros et al. (2017) suggested an average age of 5 years attributed to the timing of rapid urban growth during the economic expansion between 2009 and 2015.

Climatologically in the Manaus city the mean air temperature does not show strong seasonal variations due to the high incidence of solar radiation throughout the year. The highest temperatures are observed during the dry season, with a September monthly mean of 27.5°C, whereas the lowest temperatures prevail in the rainy season, with a monthly mean of 25.9°C in March. Rainfall in the region shows a pronounced seasonal variation, with the highest amounts in March (335.4 mm) and the lowest amounts in August (47.3 mm), with an average annual total of 2307.4 mm (Andreae et al. 2015). A typical characteristic for the central Amazon Basin is the synoptic changes between the wet and dry seasons; a good example of this is in the seasonal variations of the IntraTropical Convergence Zone (ITCZ) (Fisch et al., 1998; Wang and Fu 2007; Nobre et al., 2009). In the wet season, the Manaus plume aside, the Amazon basin is one of the cleanest continental regions on Earth (Andreae, 2007; Martin et al., 2010). In general, the preferential wind direction is from the Northeast (NE), featuring the occurrence of the breeze circulation in the same direction as the trade winds. Santos et al. (2014) observed that during the wet season, the trade winds flows more frequently from the Northeast (NE), while in the dry season they flow from the Southeast (SE). It has been suggested that these features are modulated by a land breeze circulation that is induced by the thermal gradient between Manaus City and the river water surfaces.

### 2.2 General methodology description

In this study, we applied a bottom-up approach to develop a vehicular emission inventory for the main anthropogenic emissions of an urban area: carbon monoxide (CO), nitrogen oxide ($NO_x$), methane ($CH_4$), particulate matter (PM), carbon dioxide ($CO_2$) and non-methane hydrocarbon (NMHC). Thus, given a segment of road we established an hourly traffic profile for each vehicle category. Therefore, the emissions were calculated as follows in Eq. (1):

$$E_{i,j}^{p} = \sum_{c} EF_{c}^{p} \times VTK_{c,i,j} \times 10^{-3} \tag{1},$$

where $E_{i,j}^{p}$ is the emitted mass of a pollutant $p$ on a segment of road $i$ at time $j$ (kg h$^{-1}$); $EF_{c}^{p}$ is the emission factor of pollutant $p$ by a vehicle category $c$ (g km$^{-1}$); $VTK_{c,i,j}$ is distance travelled of a vehicle category $c$ on a segment of road $i$ at time $j$ (km h$^{-1}$).



Due to the significant differences among different vehicle classifications, the emission factors were separated by the vehicle category. However, it was not possible to determine the exact age and fuel type of vehicle on a segment of road $i$ at moment $j$. Thus, we used the existing classification method established by the Brazilian Ministry of Environment by which the vehicles have been summarized into 4 classes as follows: light vehicle - LV (e.g., passenger car, taxi); motorcycle; public transportation
bus, and heavy-duty truck - HDT (e.g., truck). The grouping intends to reflect the different fuels used and fleet-based functions for private and public transport or transport of goods.

The method to calculate VKT in a given time $j$ for a segment of road $i$ is to multiply the length of a road segment ($L_i$) by the volume of traffic of category $c$ on that segment $i$ in a given time $j$, as shown by Eq. (2):

$$VTK_{c,i,j} = VT_{c,i,j} \times L_i \qquad (2).$$

It is important to note that the VKT calculation does not include travel on all Manaus roads. Small residential streets are not used in the VKT calculation; the traffic volume varies or the traffic count sample on these roadways is too small or frequently inexistent to make a reliable estimation.

### 2.2 Traffic flow, vehicle activity, vehicle density and emission factors


The fleet composition, vehicle activity, density and traffic flow were obtained from three local sources: The Transit Department of the Amazonas (Detran-AM); the Municipal Institute of Engineering and Traffic Control of Manaus (MIETCM) and the Municipal Institute of Urban Planning. For the traffic volume, a database of the MIETCM was used, obtained by manually counting vehicles on the main roads over two time periods: morning rush hour (6–9 AM) and afternoon rush hour (5–7 PM). The
counting was done over several days at each road between the years 2014-2015. Thus, for every selected road, the vehicle flow was continuously counted in 15-minute intervals during these two-hour periods.

The accuracy of the emission factors (EFs) is key and impacts the results of the vehicle emissions inventories. International EFs are hard to match perfectly since they are directly related to the characteristics such as vehicle emission control level, fuel type, vehicle age, accumulated mileage, inspection and maintenance, average speed, fraction of cold/hot starts, etc. Some studies
have quantified EFs by employing tunnel measurements (Martins et al., 2006; Ban-Weiss et al., 2010) or the remote sensing approach (Bishop et al., 2010). Despite the efforts to improve EFs, both of these approaches have significant limitations. Hudda et al. (2013) summarizes that tunnel measurements cannot provide variability showing only a central tendency. In contrast, remote sensing only allows for determining an individual vehicle's EF unless large numbers of locations are sampled in a representative manner. Therefore, in this research, we assumed the emission factors suggested by Cancelli and Dias (2014) based
on data from the Environmental Agency of São Paulo (CETESB) and the first National Atmospheric Emissions from Road Vehicles Inventory developed by Brazilian Ministry of the Environment (MMA, 2011), which were determined for Brazilian vehicles (Table 1).

### 2.3 Spatial Evaluation

To spatial distribution was been compared with values of Emission Database for Global Atmospheric Research harmonized
(EDGAR-HTAP, http://edgar.jrc.ec.europa.eu/htap, Janssens-Maenhout et al., 2012); REanalysis of the TROpospheric chemical composition over the past 40 years (RETRO, http://retro.enes.org) and the urban emissions inventory for South America (Alonso et al, 2010). Thus, version 1.5 of the PREP-CHEM-SRC system (ftp://ftp.cptec.inpe.br/brams/PREP-CHEM/PREP-CHEM-SRC-1.5.tar.gz) was utilized to standardize the emission inventories in 1km horizontal resolution. PREP-CHEM-SRC is a





comprehensive tool that aims to prepare emission fields of trace gases and aerosols for use in atmospheric-chemistry transport
models (Freitas et al., 2011).

**3 Results and discussion**

This section introduces our bottom-up vehicle emission inventory for the Manaus urban area, as resulted from applying the
methodology described above and data and information locally collected. Traffic density has been intensively monitored for a
number of years in the Manaus urban area; however, we only considered the data information for years 2014 and 2015. Figure 2
introduces the traffic density for the 258 roads for light vehicle, motorcycles, bus and truck, respectively. The thickness of the
line on the map relates to the amount of traffic. It can be seen, direction preferential for light vehicles, bus and trucks, while
motorcycles show a more homogenous distribution.

Table 2 shows a statistical summary of vehicle emissions from different categories over the study area. The totals of
emissions are 4.1, 1.0, 0.37, 0.072, 63.5 and 2.56 ton/h for $CO$, $NO_x$, $CH_4$, $PM$, $CO_2$ and $NMHC$, respectively. From these totals
the light vehicles are responsible for a large fraction of the pollutants, contributing 2.6, 0.87, 0.32, 0.03, 456 and 0.8 tons/h.

We noted that light vehicles represent about ~62.8 %, 84.7 %, 87.9 %, 45.1 %, 71.8 %, and 33.9 % and motorcycles in the
order of 32.28 %, 6.54 %, 12.06 %, 6.19 %, 14.77 %, and 8.71 %. Nevertheless, we can highlight the bus emissions which are
around ~35.74% and 45.27% for NMHC and PM, respectively (Table 3). Réquia Junior et al (2015) found for light vehicles
emissions values at Federal District - Brazil equal 68.9 % of CO, 93.6 % of CH4 and 57.9 % of CO2, and for heavy vehicles (bus
and trucks) observed values were in the order of 92.9 % and 97.4 % for NHMC and PM, respectively. In a recent study that Silva
et al. (2016) conducted from Pelotas, Brazil, results suggested that light vehicles and motorcycles contribute ~63 % and 28 % of
CO, respectively, while light vehicles, heavy vehicles (bus and trucks) and motorcycles contribute some 72 % of NMHC, 76 %
of $NO_x$ and 18 % of NMHC. Duarte et al. (2013) suggested that light vehicles in Rio de Janeiro, Brazil, contribute some 60 % of
CO, 17 of $NO_x$ and 2.9 of PM while motorcycles and heavy vehicles emit 90 % and 5.9 %, respectively. In addition, Ueda and
Tomaz (2011), using information from Campinas, Brazil, showed that light vehicles are responsible for 74 % of CO, while 61 %
of $NO_x$ and 99.9 %of PM are associated with heavy vehicles.

Figure 3 and 4 show the spatial distributions from emission inventories of CO and $NO_x$ for four vehicle categories. Although
the spatial distribution trends of the pollutants' emissions are similar, there are some differences between categories of vehicles.
Analyses of the distribution of emissions show that they are mostly higher in central zones due to the high traffic rate of
passenger cars, motorcycles, buses and trucks. This pattern was observed in the results showed in Figure 2. Of note: trucks are
not allowed to enter residential areas, which leads to a considerable level of truck-related emissions on suburban roads around
Manaus and industrial areas.

To evaluate uncertainty in the spatial distribution and total emission in the urban area of Manaus, we compared the inventory
proposed here (Local inventory - LI) with values of EDGAR-HTAP, RETRO and Alonso et al (2010). Despite the efforts to
improve the level of detail, significant uncertainties still remain because are adopted different approaches as different spatial
disaggregation methods, different resolutions, different criteria and others. For the Manaus urban area Alonso et al (2010)
suggested CO and $NO_x$ values of 23.93 and 10.16 Gg year[-1], respectively. In contrast, the present results show values of CO and
$NO_x$ around 67.34 and 32.86 Gg year[-1]. In general, for the study area, all databases utilized underestimated the total emissions of
CO and $NO_x$ emissions presented in LI by a factor of ~3. Similar results were found by Abdallah et al. (2016) for Lebanon-
Middle East. The comparison of EDGAR-HTAP, RETRO and Alonso et al. (2010) over Manaus highlights the discrepancies
between the inventories both in terms of total mass as in spatial distribution. In LI emissions we show a better distribution over



the domain, highlighting major urban roads (Figure 5). The differences between the totals and the spatial distribution may be related with the methods applied but also have a direct relationship with the size of the urban area and number of vehicles

considered in the studies. For example, for Manaus Alonso et al. (2010) utilized an area of ~265.32 km$^2$ and a total of ~581,000 vehicles; in contrast, for the inventory proposed here, we used an area of ~447 km$^2$ and a total of ~622,000 vehicles.

## 4 Conclusion

This paper introduces a detailed bottom-up vehicular emission inventory developed for Manaus, the capital of Amazonas (Brazil), based on local information and on the city scale. The estimated mobile emissions of CO, NO$_x$, CH$_4$, PM, CO$_2$ and

NMHC are approximately 4.1, 1.0, 0.37, 0.072, 63.5 and 2.56 ton/h, respectively. In agreement with previous studies over different regions, light vehicles are mainly responsible for the total emissions, accounting for approximately 62.8 %, 84.7 %, 87.9 %, 45.1 %, 71.8 %, and 33.9 % of the respective total mass of the above pollutants. Motorcycles come in second place participating with 32.3%, 6.5%, 12.1%, 6.2%, 14.8%, and 8.7% of the total mass. However, we can highlight the bus emissions which are ~35.74% and 45.27% for NMHC and PM, respectively. Our results indicate that the developed inventory reflects the

influences of standard behavior of traffic distribution per vehicle category and better distribution over the domain. Furthermore, this inventory provides more accurate information to improve the current understanding how vehicle emissions contribute to the ambient pollutant concentrations, the direct and indirect impacts on regional climate changes and more detailed scenarios for policymakers and regulatory agencies to develop strategies for controlling the vehicular emissions and, consequently, mitigate associated impacts on local and regional scales of the Amazon ecosystems.

## Code and data availability


Most of the parts that comprise this methodology are publically available. The source code of PREP-CHEM-SRC is maintained and developed National Institute of Space Research, Center for Weather Forecasting and Climate Research by the GMAI group (Group Modeling of the Atmosphere and its Interfaces). The code and most of the global and South American emission database are available upon request to the 2nd author or can be downloaded from the ftp://ftp.cptec.inpe.br/brams/PREP-CHEM/PREP-

CHEM-SRC-1.5.tar.gz. The new inventory and source code are currently being implemented within the PREP-CHEM-SRC, researchers will be able to access its code and database used in this study, as well as more recent versions, available upon request at no cost, via repository in https://doi.org/10.5281/zenodo.1245811. In addition, researcher interested in the new motor vehicle emission inventory for Manaus urban area and source code is encouraged to contact the corresponding author.

## Acknowledgments

This work is part of the PhD thesis of the first author that is under development at the Post-Graduate Program in Climate and Environment (PPG-CLIAMB) of the National Research Institute of the Amazon (INPA) and the Amazonas State University (UEA) with financial support from Coordination for the Improvement of Higher Education Personnel (CAPES). We acknowledge the support of the Central Office of the Large-Scale Biosphere Atmosphere Experiment in Amazonia (LBA), the National Institute of Research of Amazônia (INPA), National Institute of Space Research (INPE), Center for Weather

Forecasting and Climate Research (CPTEC) and the Amazonas State University. We also thank Group Modeling of the Atmosphere and its Interfaces (GMAI) - CPTEC. The first author thanks the CAPES for the grant scholarship, linked to PPG-CLIAMB. Thanks also go to Amy Houghton for their very valuable comments to improve this work.



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



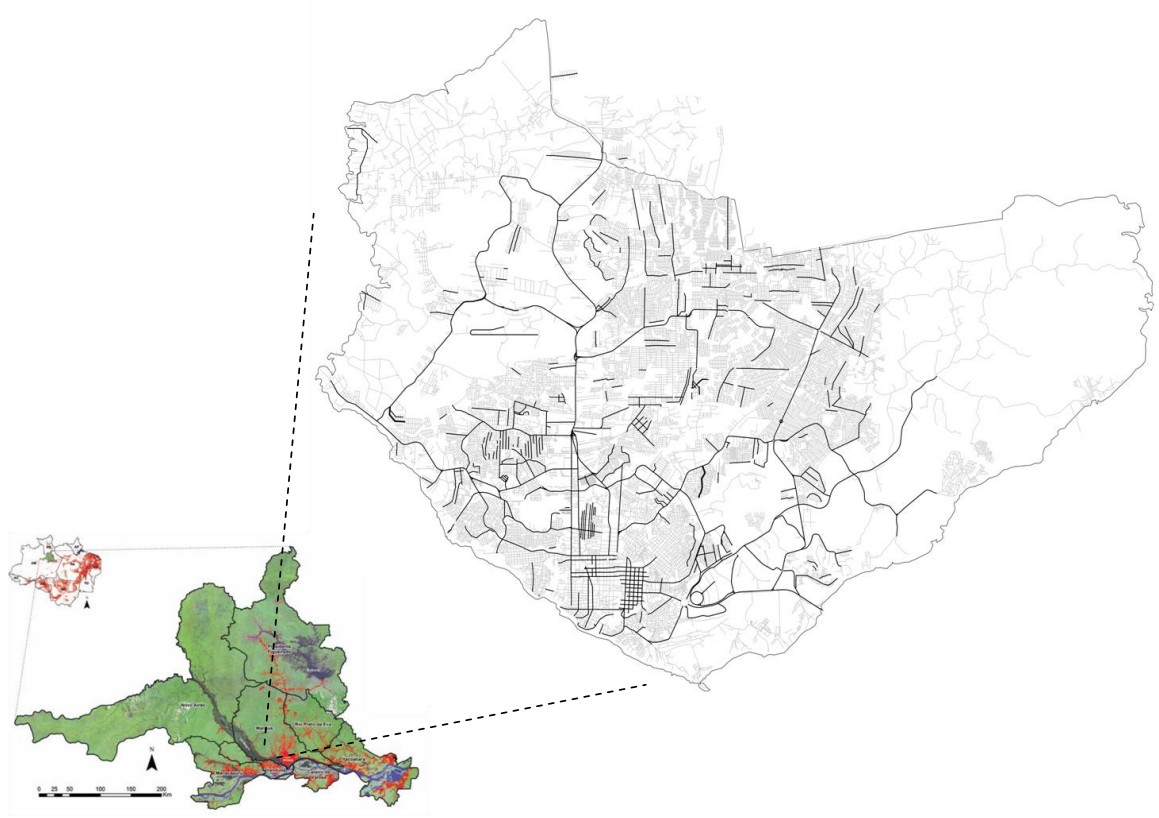

**Figure 1. Area of study. Left map represents the metropolitan region of Manaus (areas in green represent forest cover and dark lines are the municipal boundaries; area in red represents deforestation accumulated until March 2014). The map in the upper-right corner shows Manaus' urban area. The black lines represent the main roads, and grey lines are small roads.**






**Table 1. Emission factors (g km$^{-1}$) for the total number of vehicles per category (Cancelli and Dias, 2014).**

|        | light vehicles | motorcycles | bus | trucks |
|--------|---------------:|------------:|----:|-------:|
| CO     | 1.2   | 3    | 1.1 | 1    |
| NO$_x$ | 0.4   | 0.15 | 0.5 | 0.4  |
| CH$_4$ | 0.15  | 0.1  | 0   | 0    |
| MP     | 0.015 | 0.01 | 0.2 | 0.15 |
| CO$_2$ | 210   | 210  | 445 | 445  |
| NMHC   | 0.4   | 0.5  | 9   | 5    |




**Table 2. Descriptive statistics for pollutant vehicle emission (kg /h) of total number of vehicles per category for Manaus urban area using 258 roads.**

| | | CO | NO$_x$ | CH$_4$ | PM | CO$_2$ | NMHC |
|---|---|---|---|---|---|---|---|
| light vehicles | Average | 10.108 | 3.369 | 1.263 | 0.126 | 1768.815 | 3.369 |
| | standard deviation | 16.006 | 5.335 | 2.001 | 0.200 | 2801.024 | 5.335 |
| | variation coefficient | 1.584 | 1.584 | 1.584 | 1.584 | 1.584 | 1.584 |
| | Minimum | 0.138 | 0.046 | 0.017 | 0.002 | 24.234 | 0.046 |
| | Maximum | 151.006 | 50.335 | 18.876 | 1.888 | 26426.006 | 50.335 |
| | Total | 2607.739 | 869.246 | 325.967 | 32.597 | 456354.276 | 869.246 |
| | | CO | NO$_x$ | CH$_4$ | PM | CO$_2$ | NMHC |
| Motorcycles | Average | 5.196 | 0.260 | 0.173 | 0.017 | 363.714 | 0.866 |
| | standard deviation | 7.554 | 0.378 | 0.252 | 0.025 | 528.754 | 1.259 |
| | variation coefficient | 1.454 | 1.454 | 1.454 | 1.454 | 1.454 | 1.454 |
| | Minimum | 0.063 | 0.003 | 0.002 | 0.000 | 4.406 | 0.010 |
| | Maximum | 55.247 | 2.762 | 1.842 | 0.184 | 3867.309 | 9.208 |
| | Total | 1340.545 | 67.027 | 44.685 | 4.468 | 93838.161 | 223.424 |
| | | CO | NO$_x$ | CH$_4$ | PM | CO$_2$ | NMHC |
| Bus | Average | 0.550 | 0.250 | * | 0.100 | 222.525 | 4.501 |
| | standard deviation | 0.917 | 0.417 | * | 0.167 | 371.092 | 7.505 |
| | variation coefficient | 1.668 | 1.668 | * | 1.668 | 1.668 | 1.668 |
| | Minimum | 0.008 | 0.004 | * | 0.001 | 3.175 | 0.064 |
| | Maximum | 8.359 | 3.800 | * | 1.520 | 3381.649 | 68.393 |
| | Total | 141.916 | 64.507 | * | 25.803 | 57411.443 | 1161.130 |
| | | CO | NO$_x$ | CH$_4$ | PM | CO$_2$ | NMHC |
| Trucks | Average | 0.241 | 0.096 | * | 0.036 | 107.284 | 1.205 |
| | standard deviation | 0.484 | 0.194 | * | 0.073 | 215.360 | 2.420 |
| | variation coefficient | 2.007 | 2.007 | * | 2.016 | 2.007 | 2.007 |
| | Minimum | 0.002 | 0.001 | * | 0.000 | 0.809 | 0.009 |
| | Maximum | 5.616 | 2.246 | * | 0.842 | 2498.939 | 28.078 |
| | Total | 62.201 | 24.880 | * | 9.330 | 27679.313 | 311.004 |

Note: * no calculated







**Table 3. Summary of the contributions, in percentages, from air pollutant emission for each vehicle type, for Manaus urban area using 258 roads.**


| Vehicle category | CO (%) | $NO_x$ (%) | $CH_4$ (%) | PM (%) | $CO_2$ (%) | NMHC (%) |
|---|---|---|---|---|---|---|
| light vehices | 62.80 | 84.75 | 87.94 | 45.15 | 71.83 | 33.89 |
| Motorcycles | 32.28 | 6.54 | 12.06 | 6.19 | 14.77 | 8.71 |
| Bus | 3.42 | 6.29 | * | 35.74 | 9.04 | 45.27 |
| Trucks | 1.50 | 2.43 | * | 12.92 | 4.36 | 12.13 |
| Total | 100 | 100 | 100 | 100 | 100 | 100 |

Note: * no calculated





**Figure 2. The top panel shows the spatial distribution of traffic density in main roads for each vehicle category, respectively: light vehicles (a); bus (b); trucks (c); motorcycles (d). The bottom panel shows the boxplot for each vehicle category. Upper and lower whiskers represent 1.5 interquartile ranges for the period, from the 25th and 75th percentiles; outliers are represented by dots. The crosses represent sample median value and the black square is the average value.**






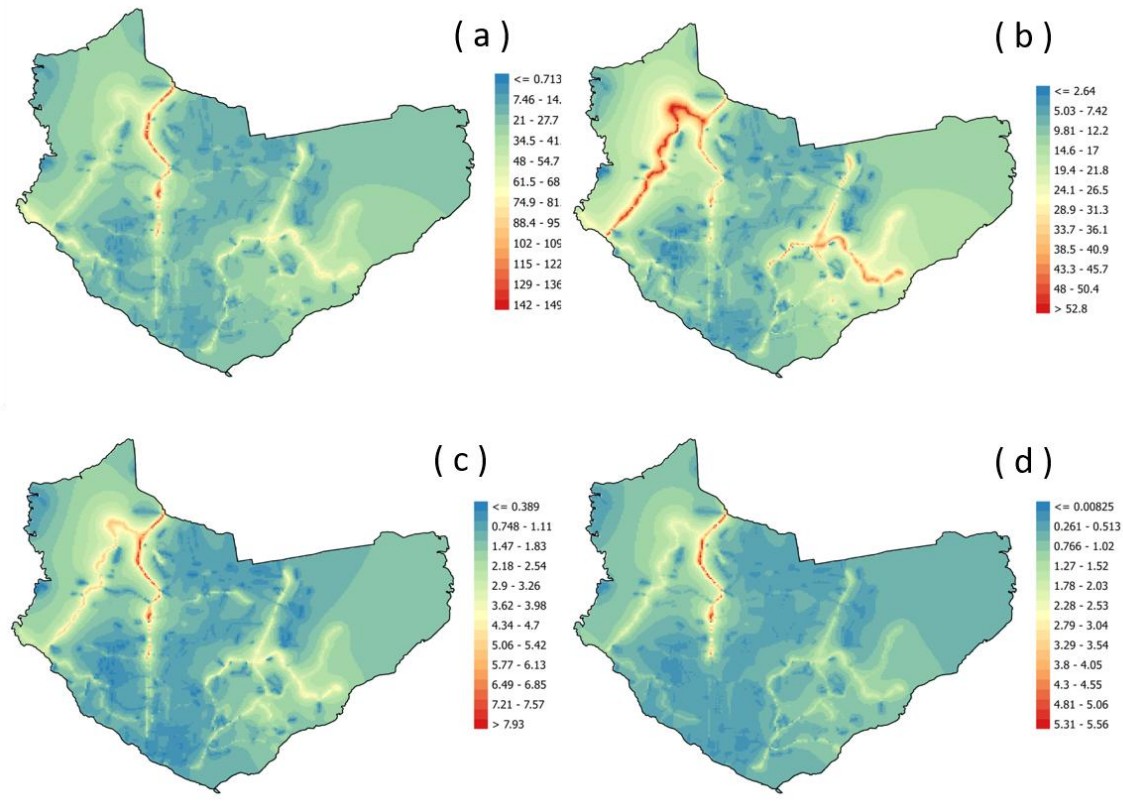

**Figure 3. Spatial distribution of CO at Manaus urban area for each vehicle category in kg h$^{-1}$, respectively: light vehicles (a); motorcycles (b); bus (c); trucks (d).**






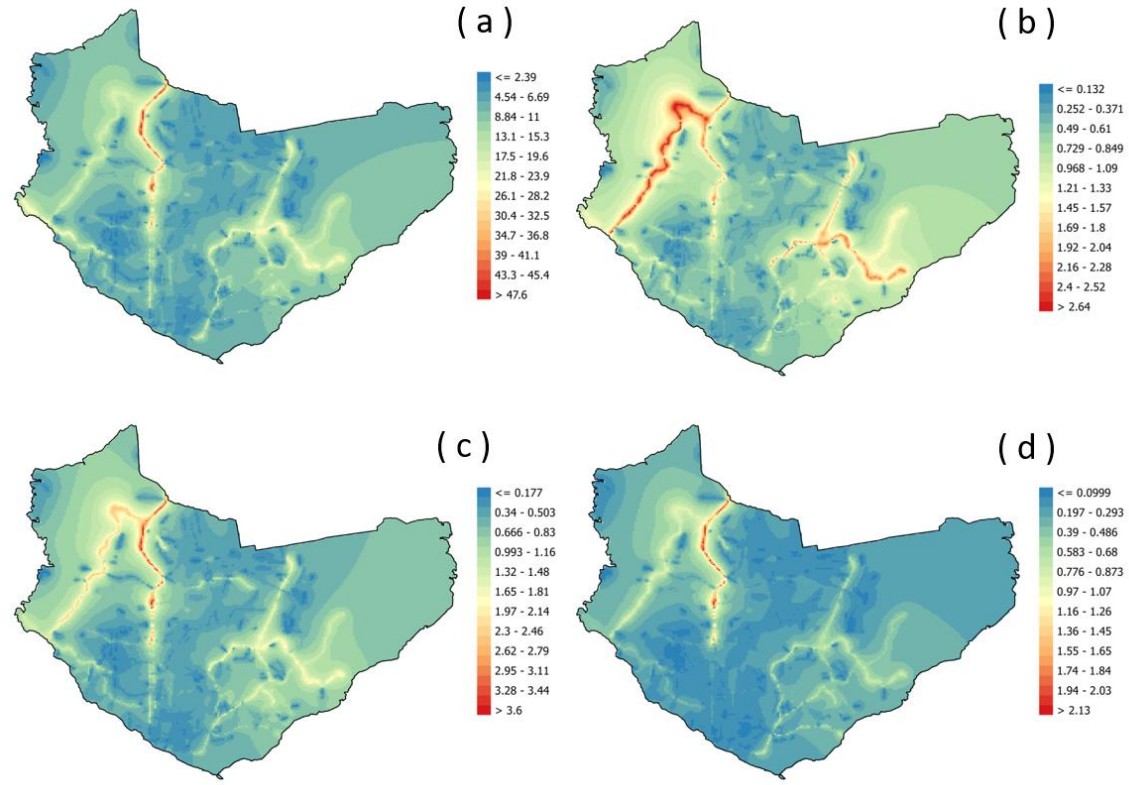

**Figure 4. Spatial distribution of NO$_x$ at Manaus urban area for each vehicle category in kg h$^{-1}$, respectively: light vehicles (a); motorcycles (b); bus (c); trucks (d).**




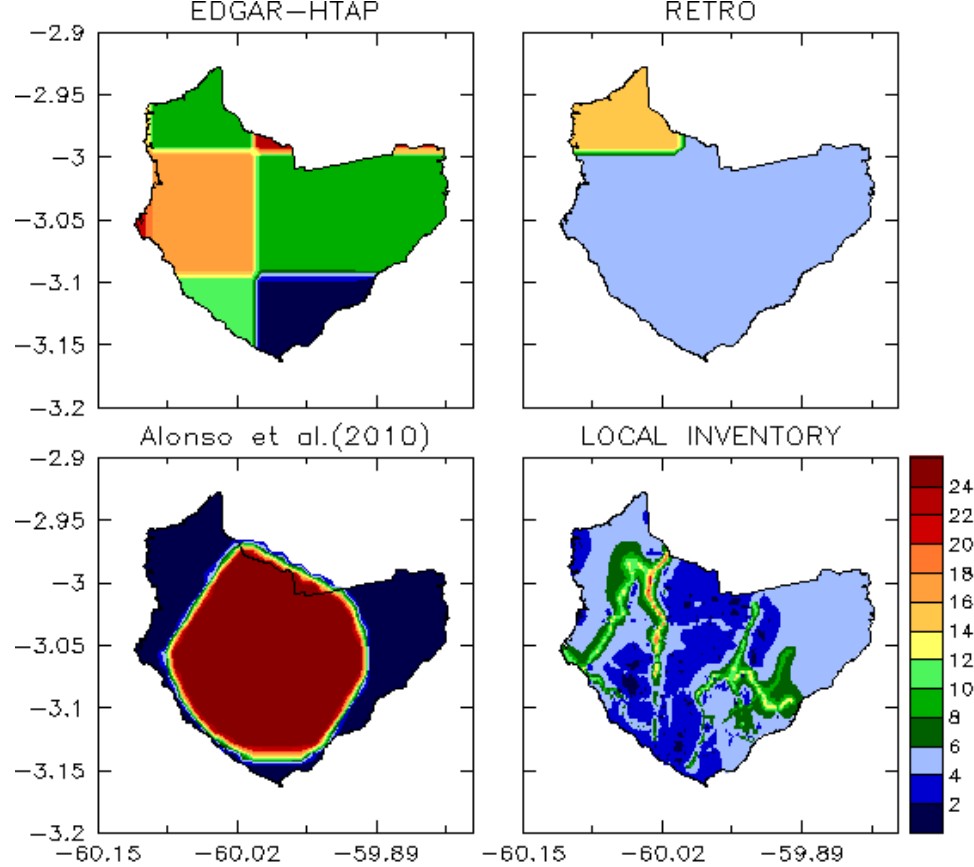

**Figure 5. Spatial distribution of CO ($10^{-6}$ kg m$^{-2}$ day$^{-1}$) at Manaus urban area.**