# Peer review of "MOVEIM v1.0: Development of a bottom-up motor vehicular emission inventories for the urban area of Manaus in central Amazon rainforest."

_Geoscientific Model Development, 2018_

## Referee Comment (RC1) · Anonymous Referee #1 · 22 Jun 2018

In term of technical for this manuscript:

strength point: use the secondary information of 258 roads in the calculation

weak point:

- Methodology used in the calculation equal to Tier 1 (just roughly estimation, don't consider on the factors that related to the emission (as engine type, etc.).

- Result not represent for the whole area (as mentioned in the manuscript, traffic volume is only from major road), it will be better if it has some explanation that mention

the coverage of the major road –> it can used to calculate the uncertainty (from the completeness of information) of this result.

- In the methodology mentioned only the traffic volume during the peak time, meaning that the result from this study represents only emission during the peak time, isn't it? No detail (unit) of traffic volume even in the figure 2. and no detail in the methodology that mentioned about the traffic volume during off-peak period.

- the figure 3 and 4 and the name under figures might be wrong. The result from figure 3 and 4 are mismatched with the traffic volume that presented in figure 2. Please check the result and the name it might be mismatched.)

―――――――――――――――――――

---

## Referee Comment (RC2) · Anonymous Referee #2 · 28 Jun 2018

1) It is important to emphasize the methodology used to extrapolate the VTK information, obtained at peak times, for the others hours (off-peak period).

2) Why not use fleet age in the categorization and emissions factors? For example, the CO EF used (1.2 g / km) is very high for the fleet age that the author mentions in the introduction ("...average age of 5 years attributed to the timing of rapid urban growth during the economic expansion between 2009 and 2015..."). This can lead to an overestimation of emissions. The author can use a Top-down approach to age distribution (From a national database like ANFAVEA) or local information from DETRAN.

3) The authors considered evaporative emissions?

4) With regard to fuel, the authors can estimate from consumption (local information) or from the national inventory (This hypothesis - top-down approach - is more subject to overestimation).

5) Please, check the NOx emission factor used to trucks (0.5 g/km). I am thinking there is an underestimation of NOx emission by heavy vehicles.

---

## Author Comment (AC1) · 6 Nov 2018

We thank the referee for his(er) insightful and very helpful comments, which contributed to improve the manuscript.

The answers to his(er) questions and comments are below:

RC: Referee's Comment

AR: Author's response

**1. RC:**

**It is important to emphasize the methodology used to extrapolate the VTK information, obtained at peak times, for the others hours (off-peak period).**

**AR:** *We thank the reviewer for this perspective and your comment. However, in front of serious limitations, our approach taken was therefore to look at VTK only at peak times. Unfortunately, in most Brazilian cities the traffic volume is observed only during at these times. Thus, even with the present limitation, we believe that this work is important for the tropical area context that still suffers serious limitations in the spatial and temporal representativeness of the urban emissions. Additionally, for vehicle emission studies and modeling, the peak hours are of higher importance than off-peak periods. In addition, the results shown here may be used in combination with tools that apply the function double gaussian distribution (e.g. PREP-CHEM-SRC; Freitas et al., 2011).*

**2. RC:**

**Why not use fleet age in the categorization and emissions factors? For example, the CO EF used (1.2 g / km) is very high for the fleet age that the author mentions in the introduction (". . .average age of 5 years attributed to the timing of rapid urban growth during the economic expansion between 2009 and 2015. . ."). This can lead to an overestimation of emissions. The author can use a Top-down**

**approach to age distribution (From a national database like ANFAVEA) or local information from DETRAN.**

**AR:** *We thank the reviewer for this perspective. However, we used the emissions factors suggested in Cancelli and Dias (2014). The emissions factors suggested by Cancelli and Dias (2014) were found used a top-down approach with based on datasets from the Environmental Agency of São Paulo (CETESB) and the first National Atmospheric Emissions from Road Vehicles Inventory developed by Brazilian Ministry of the Environment (MMA, 2011). In addition, the fleet average age of 5 years cannot be used as basis for the calculating, because the emitted mass of a specific pollutant is represent in our work as the emitted mass of the vehicle that travel a segment of a certain road and time.*

**3. RC:**

**The authors considered evaporative emissions?**

**AR:** *Unfortunately, it has not been possible to considered evaporative emissions. However, we are not aware of similar studies and measures on cities in a tropical forested region like Amazonas rain forest. But, on the other hand, we aim to further contribute in future versions the local inventory with exhaust and evaporative emissions as soon as there the deployment of the regulatory politics of evaporative emission control system in local scale for environmental agencies.*

**4. RC:**

**With regard to fuel, the authors can estimate from consumption (local information) or from the national inventory (This hypothesis - top-down approach - is more subject to overestimation).**

**AR:** *We thank the reviewer. But, the purpose of this work was to develop a vehicular emission inventory based on local vehicle activity using a bottom-up methodology. The fleet vehicular in Manaus is a specific case,*

*because many vehicles are simply transiting through to neighboring towns, thus the fuel consumption can create a false value and strong overestimation vehicular emission.*

**5. RC:**

**Please, check the $NO_x$ emission factor used to trucks (0.5 g/km). I am thinking there is an underestimation of $NO_x$ emission by heavy vehicles.**

**AR:** *We thank the reviewer for this comments. However, we check the $NO_x$ emission factor suggested in Cancelli and Dias (2014). We agreed that $NO_x$ emission factor used to trucks can make values underestimation $NO_x$ emission, but the value indicated in Cancelli and Dias (2014) consider informations for several vehicular ages. We believe that this approach constitute a good solution, manly in regions without factor emission measure.*

**References**

Cancelli, D. M. and Dias, N. L. BRevê: uma metodologia objetiva de cálculo de emissões para a frota brasileira de veículos. Eng. Sanit. Ambient., Rio de Janeiro, 19, n. spe, 13-20, 2014.

Freitas, S. R., Longo, K. M., Alonso, M. F., Pirre, M., Marecal, V., Grell, G., Stockler, R., Mello, R. F., and Sánchez Gácit a, M.: PREP-CHEM-SRC – 1.0: a preprocessor of trace gas and aerosol emission fields for regional and global atmospheric chemistry models, Geosci. Model Dev., 4, 419-433, https://doi.org/10.5194/gmd-4-419-2011, 2011.

MMA.: 1º Inventario Nacional de Emissões Atmosféricas Por Veículos Automotores Rodoviários, first ed. Ministério do Meio Ambiente, Brasília, Brasil, 2011.

---

## Author Comment (AC2) · 7 Nov 2018

Dear editor,

The authors are grateful to the reviewer's remarks and thanking him/her for very helpful comments. Please also note that was uploaded zip files with point-by-point responses to all comments and the manuscript marked-up version with check grammar, spelling and change in the manuscript. All files were uploaded in the form of a supplement.

[Figure]

Please also note the supplement to this comment:
https://www.geosci-model-dev-discuss.net/gmd-2018-81/gmd-2018-81-AC2-
supplement.zip

———————————————————